# Changes in Yield-Related Traits, Phytochemical Composition, and Antioxidant Activity of Pepper (*Capsicum annuum*) Depending on Its Variety, Fruit Position, and Ripening Stage

**DOI:** 10.3390/foods12213948

**Published:** 2023-10-29

**Authors:** Karima Lahbib, Fethi Bnejdi, Gaetano Pandino, Sara Lombardo, Mohamed El-Gazzah, Safia El-Bok, Samia Dabbou

**Affiliations:** 1Laboratory of Biodiversity, Biotechnology, and Climate Changes, Faculty of Sciences of Tunis El Manar El Manar II, Tunis 2092, Tunisia; fethi.benejdi@issacm.u-sousse.tn (F.B.); mohamed.elgazzah@fst.rnu.tn (M.E.-G.); safia.elbok@fst.utm.tn (S.E.-B.); 2Department of Agriculture, Food and Environment, University of Catania, via Valdisavoia 5, 95123 Catania, Italy; g.pandino@unict.it (G.P.); sara.lombardo@unict.it (S.L.); 3Faculty of Dental Medicine, University of Monastir, Avicenne Street, Monastir 5019, Tunisia; 4Unit of Bioactive and Natural Substances and Biotechnology UR17ES49, Faculty of Dental Medicine of Monastir, University of Monastir, Avicenne Street, Monastir 5019, Tunisia

**Keywords:** pepper, upper parts, maturity stage, total phenolic content, antioxidant activity, PCA

## Abstract

The relationship between fruit position, ripening stage, and variety has not been well studied in pepper plants. To understand the interaction of these factors, a diversity of phytochemical traits as well as antioxidant activity were investigated with agronomic traits in eleven hot pepper varieties collected from the upper and lower parts of the plant and harvested at three maturity stages (green, orange, and red). Capsaicin content (CAP) showed a relatively high genetic effect; on the contrary, total phenolic content (TPC), total flavonoid content (TFC), and antioxidant activity were more affected by the ripening stage and fruit position. The CAP values ranged from 0.29 (‘FKbM’) to 0.77 (‘Bka’) mg CAP equivalents g^−1^ DW. The ripening stage was the predominant factor for TPC, TFC, DPPH, and FRAP. There was no significant interaction between A × FP, A × RS, and FP × RS for all agro-morphological fruit traits. Variety, fruit position, and ripening stage effects are more significant than all interactions calculated. Lower fruit positions in all samples showed a maximum fruit size, whereas phytochemical traits and yield per plant were relevant in the upper parts, and Phytochemical traits and yield per plant were significantly correlated. From PCA and cluster analysis, all varieties showed the highest biochemical and antioxidant levels with moderate fruit size, except the ‘Bel’ variety that showed the smallest fruit traits with high yields, and the ‘FKbM’ and ‘FKbK’ varieties that showed the highest fruit size but low yields. This study supplies information to identify interesting cultivars with considerable levels of bioactive and phytochemical metabolites, which is useful for breeding programs of novel varieties.

## 1. Introduction

The fruit of pepper is recognizable from other plants due to its notable biochemical composition. The presence of bioactive components in fruit composition, including capsicinoids and phenolic compounds, result in its particular flavor with high nutritional benefits [1,2]. A wide range of variability characterizes the amounts and profiles of these compounds as a consequence of genetic and environmental influences [3,4,5,6,7,8,9].

The impact of postharvest factors and their interactions on pepper production can have a significant effect on its quality, yield, and phytochemical composition. Preharvest factors include agricultural and environmental conditions such as temperature, rainfall, maturity stages, diseases, and fertilization.

In addition, fruit position influences fruit characteristics, as observed by Estrada et al. [10], who showed that capsaicinoids follow an increasing gradient along the stem. Previous studies have also shown that the commercial and nutritional values of fruits and vegetables are significantly affected by their position in the plant, as observed in tomatoes [11,12], apple fruit [13], olive [14] pomegranate fruit [15], and globe artichoke [16,17]. The combined effect of environmental factors on fruit position in the plant has not yet been investigated in pepper, especially in fruit ripening stages. Indeed, changes in phenolic composition and antioxidant activity occur during pepper fruit development [18,19,20,21]. Therefore, it is recommended that pepper fruits should be harvested at their suitable maturity stage, accumulating high amounts of nutraceutical compounds. Lahbib et al. [21] found their highest amounts at the latest stage of ripening, especially for total phenolic content and antioxidant activity, while a high total flavonoid content was detected at the early harvest stage [21].

Currently, researchers are attempting to ascertain relationships among various environmental factors, such as temperature, incident light, maturation, or fruit position, affecting the productivity and composition of pepper plants [5,10,19,20,22,23,24]. However, there is a lack of understanding of how fruit position in the plant of pepper can affect fruit composition during the harvest season to realize eventual interactions between these factors. An integrated approach considering several factors is essential for optimizing production. Therefore, the target of this work was to evaluate the combined effect of variety, fruit position, and ripening stage on yield-related traits and phytochemical composition of Tunisian chili peppers.

## 2. Materials and Methods

### 2.1. Plant Material and Sample Processing

Seedlings of eleven Tunisian varieties of chili pepper (Baklouti Chébika, ‘BaklC’; ‘Beldi’, ‘Bel’; ‘Chaabani, ‘Chba’; ‘Sisseb Chébika, ‘SisC’; ‘Bkalti, ‘Bka’; ‘Knaiss, ‘Kna’; ‘Baklouti Sbikha, ‘BaklS’; ‘SissebSbikha, ‘SisS’; ‘Fort Menzel Temim, ‘FkbM’; ‘Fort de Korba’’ ‘FkbK’; and ‘Corne de Gazelle, ‘CGaz’) were planted in the experimental field of the Faculty of Science of Tunis, El Manar University (humidity: 73%; annual mean relative temperature: 20.1 °C; annual mean total rainfall: 415 mm), during spring of 30 March 2017 using a randomized complete block design with three replications in a drip-irrigated spacing of 4 m × 3 m. Fruits of different varieties were collected and evaluated at three different maturity stages, green, orange, and red ripe fruits, hereafter indicated as ‘S1’ (90 days after planting), ‘S2’ (125 days after planting), and ‘S3’ (150 days after planting), respectively. Pepper fruits were harvested, and the mean temperatures were calculated from the measurements taken before each harvest (Table 1).

The fruits of chili peppers were harvested from 15 July to 15 September 2017 from two different plant positions, the lower and upper parts, hereafter referred to as PI and PII, respectively. Upper fruits are those located higher up on the plant at the 6th foliar stage, closer to the principal stem or trunk, while lower fruits are positioned lower on the plant, closer to the ground at the second foliar stage. Plants of chili pepper were fertilized and controlled with pest treatments undertaken according to the recommendations authorized by the Department of Agriculture, Ministry of Agriculture and Cooperatives. Three plants were chosen, and five fruits per plant were assessed with regard to fruit position within a particular plant, variety, and maturity stage (30 samples per variety). Only fruits free from visible defects, diseases, or damage were used. After picking fruits, their agro-morphological traits were assessed, and then, their chemical evaluation in the laboratory was immediately carried out.

### 2.2. Agro-Morphological Characterization

Biometric traits assessed were fruit fresh weight (FW, expressed as g) measured with the FA-G series electromagnetic balance; fruit length (FL, expressed as cm); and fruit thickness (Fth, expressed as mm) determined with a digital caliper. The harvest was recorded as the number of fruits per plant.

### 2.3. Phytochemical Analysis

Capsaicin content (CAP), total phenolic content (TPC), total flavonoid content (TFC), and antioxidant activity analysis using 1,1-diphenyl-2-picryl hydrazyl (DPPH) scavenging and ferric reducing power (FRAP) assay were performed. Analyses were performed on fruits collected from both PI and PII of each pepper cultivar at the three maturity stages. All samples were analyzed in triplicate.

#### 2.3.1. Capsaicin Determination

CAP content in the eleven Tunisian varieties of chili pepper was determined according to the Sadasivam and Manikkam [25] method. Briefly, CAP allows the reduction of phosphomolybdic acid to lower acids of molybdenum. The resulting blue color intensity is directly proportional to the concentration of CAP. An aliquot of 10 mL of acetone was used for the extraction of two grams of sample. The obtained clear supernatant (1 mL) was then evaporated to dryness in a hot water bath and centrifuged to remove the floating debris. Then, to dissolve the obtained residue, 0.4 mL of NaOH and 3 mL of 3% phosphomolybdic acid were added, and the mix was kept aside for an hour. Then, the mix was centrifuged for 15 min at 5000 rpm, and the absorbance from the clear blue solution was read at 650 nm, against a blank. The CAP content calculated from the standard curve was estimated as mg of CAP equivalents g^−1^ of dry weight (DW). A UV-Vis spectrophotometer (BioSpectrometer^®^ Series, Hamburg, Germany) was used to measure the absorbance of the clear blue solution obtained.

#### 2.3.2. Total Phenolic Content

The TPC was estimated spectrophotometrically using the modified Folin–Ciocalteu colorimetric method proposed by Ghasemnezhad et al. [19]. Each sample (1 g) was extracted with 10 mL methanol. An aliquot of 125 μL of the methanolic supernatant extract was added to 375 μL of distilled water and 2.5 mL of 10% Folin–Ciocalteu. After standing for 6 min, 2 mL of 7.5% sodium carbonate solution was added, and all the samples were cooled to room temperature before reading the absorbance at 765 nm. TPC, calculated from the standard curve, was estimated as mg gallic acid equivalent (GAE) g^−1^ dry weight (DW).

#### 2.3.3. Total Flavonoid Content

The TFC was estimated using the protocol by Um and Kim [26,27]. A mix of 30 μL of sodium nitrite (5%) and 100 μL of extract were added to 30 μL of aluminum chloride (10%) after 5 min, and the obtained solution was incubated at room temperature for 6 min. An aliquot of 100 μL of 1 N NaOH and 25 μL of distilled water were added at the latest to the solution. The absorbance of the obtained solution was read at 415 nm, and the TFC was expressed as mg naringin equivalent (NAE) g^−1^ DW.

#### 2.3.4. DPPH-Scavenging Assay

The free radical-scavenging activity of each sample extract was determined based on the scavenging activity of DPPH as described previously [28]. After 30 min, the absorbance of each sample was estimated at 515 nm. The percent scavenging was calculated with the following equation: % DPPH scav = (Acont − Asamp) × 100/Acont, where Acont is the percentage of the absorbance, in the absence of pepper extract, and Asamp is the absorbance of the sample extract.

#### 2.3.5. Ferric Reducing Power Assay

The FRAP assay was performed as described by Benzie and Strain [29]. FRAP reagent was prepared as follows: 20 mmoL L^−1^ ferric chloride was added to 10 mmoL L^−1^ of 2,4,6-tripyridyl-S-triazine (TPTZ) in 40 mmoL L^−1^ of HCl and 300 mmoL L^−1^ of sodium acetate buffer (pH 3.6) in a ratio of 1:1:10 (*v*/*v*/*v*). An aliquot of 3 mL of the FRAP reagent was added to 100 mL of the extract. The obtained solution was mixed simultaneously and kept at room temperature for 4 min. The absorbance was estimated at 593 nm. The total antioxidant capacity was calculated as millimoles of Trolox equivalents (mmol TE) g^−1^ extract.

### 2.4. Statistical Analysis

All the traits studied were examined with ANOVA analysis using a 95% confidence interval, and a comparison with Duncan’s test and correlation analysis to reveal associations among the traits were performed. Principal component analysis (PCA) and hierarchical cluster analysis (HCA) were performed to acquire a general overview of variability among cultivars. All statistical analyses were performed using XLSTAT (2020) for Windows (Addinsoft, NewYork, NY, USA).

## 3. Results and Discussion

A comparison of the fruit agro-morphologic traits and the phytochemical evaluation were performed on pepper fruits harvested from PI and PII of each variety at three ripening stages (‘S1’, ‘S2’, and ‘S3’). All obtained results are shown in Table 2, Table 3, Table 4 and Table 5.

### 3.1. Variety Effect

Table 2 shows that the variety was the important source of variation for FW, FL, and CAP compared to the ripening stage and the fruit position (Table 2).

*Agro-morphological traits*. ANOVA analyses performed among the varieties for agro-morphological traits, including FW, FL, Fth, and total yield, were reported in Table 1. In particular, ‘FKbK’ followed by ‘FKbM’ and ‘Chba’ showed a higher average FW of 11.92, 11.32, and 11.14 g, respectively (Table 3). The highest total yield was detected in ‘Bel’ (532 g plant^−1^), while ‘FkbK’ and ‘FkbM’ exhibited the lowest total yield values (158 and 183 g plant^−1^, respectively). Tripodi et al. [2,4] showed a greater variability in fruit weight with a coefficient of variation (CV) value of above 70% compared to the total yield, revealing a CV lower than 35%. ‘FkbK’ and ‘FkbM’ varieties recorded the highest Fth (2.83 and 2.37 mm). Our results were in agreement with those of other authors [4,9,30,31,32] that showed a great variability within pepper germplasm based on its agro-morphological traits.

*Phytochemical traits*. Significant differences were observed among the varieties studied for CAP content (Table 2). Differences calculated among the eleven varieties were statistically significant (*p* ≤ 0.001). The CAP values ranged from 0.29 (‘FKbM’) to 0.77 (‘Bka’) mg CAP equivalents g^−1^ DW (Table 3), which were significantly higher than those reported by Materska and Perucka [33]. In the literature, CAP content is strongly dependent on genotype [7] and is attributed to the fruit part, as shown previously [34,35].

The TPC ranged between 6.18 (‘Bak lC’) and 3.74 (‘Kna’) mg CAP equivalents g^−1^ DW. All results provided in Table 3 were within the range of those reported by other authors [7,33,36,37]. TPC showed a lower genotypic effect in comparison with maturity stages and fruit position effects (Table 2).

Regarding the TF content, the values ranged from 0.26 (‘Bka’) to 0.39 mg NAE g^−1^ DW (‘BaklC’; ‘Chba’) (Table 3). Angel et al. [37] showed that this content ranged from 25.38 ± 3.44 to 60.36 ± 9.94 mg QE 100 g^−1^ FW in *Capsicum annuum* L. cultivars. Furthermore, this content varied from 17.17 to 85.49 mg QE 100 g^−1^ FW as observed by Howard et al. [6] in *Capsicum annuum* L. cultivars.

The results of antioxidant activity are shown in Table 3. Regarding the DPPH assay, ‘BaKlS’ and ‘Chba’ extracts presented the highest values (86.3 and 86.2% DPPH scav, respectively), while ‘SisS’, ‘Bka’, and ‘Kna’ showed the lowest values (49.9, 57.2, and 58.3% DPPH scav, respectively). Our results exhibited higher values than those observed by Materska and Perucka [33] and Angel et al. [37] who reported 15 to 77% and 8.45 to 83.44% radical scavenging activity in hot pepper, respectively. On the other hand, levels of FRAP assays observed in this study varied from 19.60 to 26.27 mmol g^−1^ (Table 3). According to Sim and Sil [38], the concentrations of reducing substances like phenolics were the source of this variation.

### 3.2. Fruit Position Effect

According to ANOVA, the fruit position represented the principal factor for Y and Fth (Table 2).

*Agro-morphological traits*: The fruits collected from the PI (lower plant part) showed the highest values of all agro-morphological traits under study (Table 3). The higher fruit dimensions collected from PI were probably linked to the rate of assimilation, translocation, and sink strength, which are higher in primary fruits receiving photoassimilates [39,40]. On the other hand, the total yield per plant increased by 30% for fruits collected from PI to PII (upper plant part).

*Phytochemical traits***:** From Table 3, fruits collected from PII reported the highest values of TPC, TFC, DPPH, and FRAP, which received more sun exposure than those in the lower layers. In particular, the increase in TP, TF, DPPH, and FRAP in PII in comparison with PI was up to 29.8, 8.0, 4.9, and 9.5%, respectively. Conversely, the CAP content was usually greater in the fruits of PI with respect to those of PII.

The lower content of TPC and TFC at PI may be associated with the higher concentrations of primary metabolites in this position, and this could slow down the synthesis of secondary metabolites such as phenolic compounds as the primary metabolites could convert first into capsaicin before forming other phenolics. The conversion of primary metabolites to capsaicin probably needs a lower light exposure. Moreover, capsaicin and phenolic pathways were concomitant since they had the same precursors [10]. Overall, better light conditions may induce the synthesis of phenolics in the sun-exposed fruit peppers. In apple cultivars, the more sun-exposed parts of the canopy were the upper positions in the canopy and exhibited better quality parameters, above all the phytochemical ones [13,41].

This is the onset report on the variation in phytochemicals and antioxidant amounts according to fruit positions on the pepper plant. Prospective studies need to investigate and elucidate the mechanism involved in the light induction pathway.

### 3.3. Ripening Stage Effect

As reported in Table 2, the ripening stage was the predominant factor for TPC, TFC, DPPH, and FRAP.

*Agro-morphological traits*: The fruit dimensions increased slightly when passing from ‘S3’ to ‘S1’ (Table 3), probably due to a slow natural assimilation transfer from leaves to fruits during the maturation process until the final red stage (‘S3’). The plant probably achieved adequate fruit development without going beyond the orange stage (‘S2’) when the accumulation of photoassimilates in the mesocarp reaches its maximum.

*Phytochemical traits*: In the ripening stage, TPC and antioxidant activity (DPPH and FRAP) increased significantly (Table 3). These values indicated the occurrence of the antioxidant precursors at a more advanced ripening stage, which allowed the formation of different antioxidant compounds with different levels of antioxidant activity such as phenolics [42]. Our data are in line with the findings of Hamed et al. [43] who reported the highest antioxidant values in the red stage. However, other studies on chili pepper demonstrated that both phytochemical level and radical-scavenging activity increase as the fruit matures [44,45]. Bhandari et al. [46] showed that different phytonutrients exhibited different levels of harvesting with a time-dependent variation; capsaicinoid showed the highest coefficient of variation (CV, 55.44%), total flavonoid content (15.97%), and vitamin C (10.57%). Total vitamin E content, phytosterols, fatty acids, and the proximate nutrients were relatively constant, with low CVs (<10.0%).

The TFC decreased significantly by 26.2% from ‘S3’ to ‘S1’ ripening stage (Table 3). Our results corroborated previous findings [6,19] which indicated that TFC decreased in ripe pepper fruit and acted as a chlorophyll photoprotector at the early developmental stages and disappeared at the latest growth stages [33]. Slight differences were observed in CAP content according to maturity stages. Its highest levels were observed at the earliest stage. This indicates that maturity moderately influences the capsaicin accumulation (Table 3).

### 3.4. Interactions Effect

There was no significant interaction between A × FP, A × RS, and FP × RS for all agro-morphological fruit traits. Interactions in relation to phytochemical traits were more significant. Fruit position and ripening stage played an important role, according to phytochemical levels rather than agro-morphological traits. Variety, fruit position, and ripening stage effects dominated all interactions (Table 2).

#### 3.4.1. Variety × Fruit Position

‘Variety × fruit position’ interaction may reflect how different varieties manage their resources such as water, nutrients, and energy to support their fruit growth and ripening.

This interaction was significant for yield per plant and all phytochemical traits except CAP content (Table 2). In contrast, it was not significant for all agro-morphological fruit traits. This suggests that fruit position plays a principal role, according to phytochemical levels. In most cases (Appendix A), higher values of yield, total phenolic content, and total flavonoid content were observed in the upper positions of all cultivars. These higher values could be related to the higher light radiation reaching the upper part of the plant, as suggested by several authors [10,47,48,49]. With respect to the bioactive compounds, fruit position influenced the variation in total phenolic and flavonoid contents compared to capsaicin content and morphological traits.

#### 3.4.2. Variety × Ripening Stage

The ‘Variety × maturity stage’ interaction is significant as it shows differences in how quickly varieties reach maturity and produce fruit.

Significant interactions were registered for all traits except for CAP content for which little interaction was observed (Table 2). According to variety × ripening stage interaction, BaklC, Chba, BaklS, and FKbM showed higher values for all biochemical characteristics except CAP content (Table 4).

Variations according to the ripening stage were reported to occur in parallel to fruit development according to the availability of photoassimilates in the plant [11]. In addition to photoassimilate availability, which influences fruit plant growth and its phytochemical composition, certain environmental stressors including temperatures, shading in greenhouse environments, and macronutrient application rates such as calcium and nitrogen applications have been reported as factors leading to yield decreases [3,38].

Overall, our results corroborate those of Tripoli et al. [4,9], indicating that the genotype was the principal source of variation for most of the agronomical and morphological traits, which were highly heritable and regulated by additive genes with little influence from the environment.

In this way, Tripodi et al. [2] studied the effects and interactions on pepper genotypes cultivated in both locations and found that the genotype effect was prevalent, showing about 90% of the total variation despite Genotype × Environment interaction being significant. This was also in accordance with previous studies [30,31,32,50], showing the principal role of genotype in capsaicinoid variation in several *Capsicum* spp. except for the nor-dihydrocapsaicin compound [51]. However, this is not in agreement with [52,53] trials, which highlighted the capsaicin variation based on environmental factors by particularly using the varieties of *C. chinense*.

#### 3.4.3. Fruit Position × Ripening Stage

The ripening stage influences the taste quality of the fruit. The position of the fruit on the plant can affect how the plant allocates its resources for fruit growth as well as the number of fruits produced by the plant. When both factors interact significantly, this indicates that the quality of the fruit depends largely on its maturity stage and its position on the plant.

This interaction has no influence on fruit yield and yield-related traits, suggesting no combined effect on the morphological fruit development and fruit yield (Table 2). Biochemicals except capsaicin and the antioxidant capacity were rather more responsive to the ripening stage and fruit position, particularly yielding primary and morphological fruit traits (Table 2). At PI, the TPC values increased from 2.93 to 4.56 mg GAE g^−1^ DW, respectively, from S1 to S3, while the observed DPPH level was 58.37 to 74.77% from S1 to S3. On the other hand, at PII, the TPC increased from 3.53 to 8.14 mg GAE g^−1^ DW, whereas the DPPH level increased from 64.21 to 84.46% from S1 to S3 (Table 5). These effects and interactions were substantial and are still hard to comprehend even now [4,9].

### 3.5. Correlations and Cluster Analysis

From the correlation matrix performed at both positions (PI and PII), strong and significant positive correlations were observed for agro-morphological traits along with CAP (Table 6). The TPC was positively correlated with the antioxidant activity at both positions. Agro-morphological traits were strongly and negatively correlated with the fruit yield. Moreover, phytochemicals and morpho-agronomic factors exhibited no differences in the coefficients of correlations at both positions. Our observations from the correlation matrix suggested the possibility of obtaining higher values of TPC and TFC by selecting smaller fruits.

The PCA obtained based on the agro-morphological and phytochemical traits collected at PI and PII has explained 78.0% of the total variance (Figure 1). Yield-related traits at PII and PI plant positions were the basic factors identifying the varieties on the first component. Yield traits associated with ‘Bel’ and ‘Kna’ and yield-related traits associated with ‘FKbM’ and ‘FKbK’ were oppositely located on the biplot. On the other hand, phytochemical traits were the essential traits discriminating all the remaining varieties associated with the second component. Particularly, ‘BaklC’, ‘Chaa’, and ‘BaklS’ were correlated with TPC, FRAP, DPPH, and FL, whereas ‘CGaz’, ‘SisS’, ‘Kna’, and ‘SisC’ were associated with the CAP content. The cluster analysis showed three groups (Figure 2). ‘Bel’ cultivar formed a separate group with the smallest fruit traits and high yields. ‘FKbM’ and ‘FKbK’ developed the second group with the highest fruit size with a low yield, and all the remaining cultivars were included in the third group, showing the highest biochemical and antioxidant levels as well as a moderate fruit size.

## 4. Conclusions

The present study provides meaningful insights into agro-morphological and phytochemical variations among eleven Tunisian chili pepper varieties, assessed in two plant positions (PI and PII) at three different ripening stages (‘S1’, ‘S2’, and ‘S3’). The CAP content was less influenced by the ripening stage and fruit position, while the TPC, TFC, and antioxidant activity were more responsive to the fruit position and ripening stage. The latter resulted in preponderance with respect to both variety and fruit plant position for the TPC, TFC, and antioxidant activity. On the other hand, the fruit position was predominant only for the Fth and yield, while variety was for both the FW and FL. Overall, our results indicated that the genotype was a dominant source of variation for most of the morphological traits, and the fruit yield was less influenced. On the other hand, phytochemical content, except that of capsaicin, was more affected by environmental conditions.

The higher average temperature and light levels intercepted by PII at ‘S3’ provided a cultivar with high levels of bioactive compounds. Thus, this study on the effects of variety, ripening stage, and fruit position and their interactions provided the possibility for breeders to choose adapted cultivars based on consumers’ preferences in terms of pungency and fruit dimensions, but rarely in terms of other phytochemical compounds related to fruit quality.

## Figures and Tables

**Figure 1 foods-12-03948-f001:**
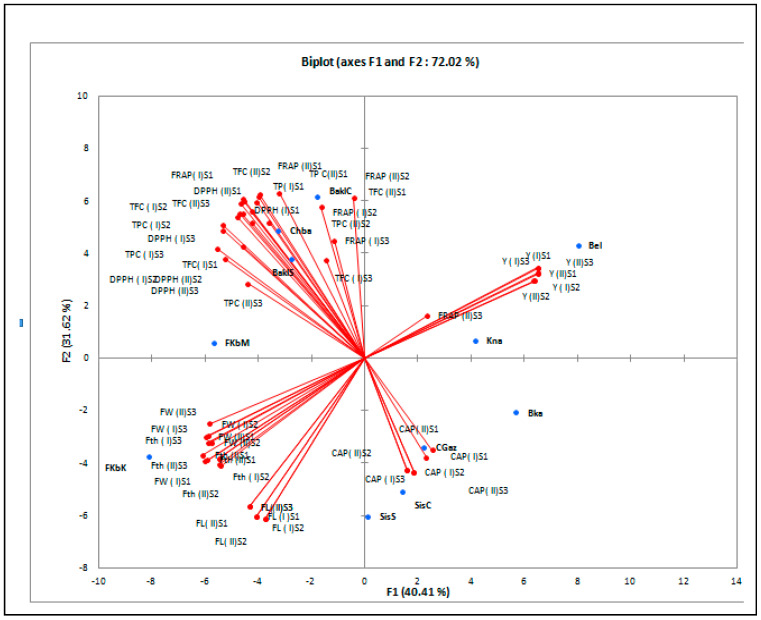
Principal components analysis (scores and loading plots, biplot (Axis1–Axis2)) based on agro-morphological and phytochemical traits analyzed according to the ripening stage (S1: green ripening stage; S2: orange ripening stage; and S3: red ripening stage) and the fruit position (lower fruit position plant (PI) and upper fruit position plant (PII)) of eleven Tunisian pepper cultivars. FW: fruit weight; FL: fruit length; Fth: fruit thickness; Y: yield; CAP: capsaicin, TPC: total phenolic content; TFC: total flavonoid content; DPPH: 1,1-diphenyl-2-picryl hydrazyl; FRAP: ferric reducing power assay; BaklC: Baklouti Chébika; Bel: Beldi; Chba: Chaabani; SisC: Sisseb Chébika; Bka: Bkalti; Kna: knaiss; BaklS: Baklouti Sbikha; SisS: Sisseb Sbikha; FkbM: Fort Menzeltemim; FkbK: Fort de korba; and CGaz: Corne de Gazelle.

**Figure 2 foods-12-03948-f002:**
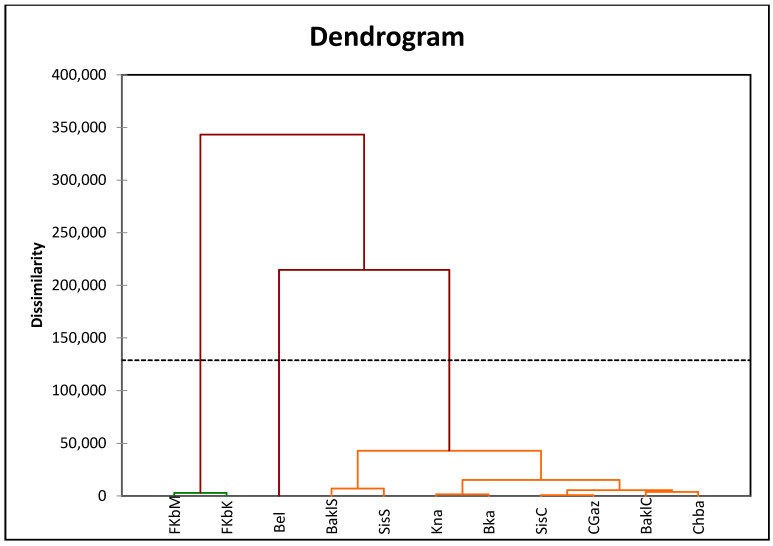
Cluster analysis based on agro-morphological and phytochemical traits analyzed according to the ripening stage and the fruit position of eleven Tunisian pepper cultivars. BaklC: Baklouti Chébika; Bel: Beldi; Chba: Chaabani; SisC: Sisseb Chébika; Bka: Bkalti; Kna: knaiss; BaklS: Baklouti Sbikha; SisS: Sisseb Sbikha; FkbM: Fort Menzeltemim; FkbK: Fort de korba; and CGaz: Corne de Gazelle. The same color lines designed varieties belonging to the same cluster.

**Table 1 foods-12-03948-t001:** Air temperature, fruit maturity, and color of the eleven Tunisian chili pepper varieties at each harvest time.

Harvest Stage	Air Temperature at Harvest (°C)	Horticultural Maturity	Fruit Color
S1 (90 days after planting)	31.4	Ripen	green
S2 (125 days after planting)	34.1	Middle Ripen	orange
S3 (150 days after planting)	36.2	Fully Ripen	red

**Table 2 foods-12-03948-t002:** The effects and the interaction of variety, fruit position, and ripening stage on the agro-morphological and phytochemical traits in eleven Tunisian chili pepper varieties. In brackets, the mean square as a percentage of total effects resulting from the analysis of variance is provided.

	FW (g)	FL (cm)	Fth (mm)	Y (g plant^−1^)	CAP (mg^−1^ DW)	TPC (mg GAE g^−1^ DW)	TFC (mgNAEg^−1^ DW)	DPPH (%)	FRAP (mmol g^−1^)
variety (A)	**	**	*	**	**	**	**	**	**
	(78%)	(68.69%)	(3.30%)	(21.25%)	(52.6%)	(4.44%)	(13.24%)	(32.48%)	(8.62%)
Fruit position (FP)	**	**	**	**	**	**	**	**	**
	(17%)	(30.75%)	(92%)	(74.18%)	(45.5%)	(39.93%)	(14.28%)	(6.18%)	(17.09%)
Ripening stage (RS)	**	*	*	ns	ns	**	**	**	**
	(1.31%)	(0.33%)	(0.07%)	(0.09%)	(0.39%)	(45.24%)	(63.41%)	(56.53%)	(71.09%)
A × FP	ns	ns	ns	*	ns	*	**	**	*
	(0.002%)	(0.00002%)	(0.0002%)	(0.21%)	(0.034%)	(0.12%)	(3.48%)	(0.33%)	(0.18%)
A × RS	ns	ns	*	ns	ns	**	*	**	**
	(0.96%)	(0.21%)	(2.45%)	(0.15%)	(0.15%)	(0.27%)	(0.34%)	(2.01%)	(2.51%)
FP × RS	ns	ns	ns	ns	ns	**	**	*	*
	(0.06%)	(0.00001%)	(0.0001%)	(0.006%)	(0.01%)	(13.91%)	(5.22%)	(0.06%)	(0.35%)

FW: fruit weight; FL: fruit length; Fth: fruit thickness; Y: yield; CAP: capsaicin; TPC: total phenolic content; TFC: total flavonoid content; DPPH: 1,1-diphenyl-2-picryl hydrazyl; FRAP: ferric reducing power assay; ** indicates significance at *p* ≤ 0.01, * indicates significance at *p* ≤ 0.05, and ns, not significant.

**Table 3 foods-12-03948-t003:** Duncan’s multiple range test according to variety, fruit position, ripening stage on agro-morphological and phytochemical traits and antioxidant activity for the studied traits in eleven Tunisian chili pepper varieties.

	FW (g)	FL (cm)	Fth (mm)	Y (g plant^−1^)	CAP (mg CAP g^−1^ DW)	TP (mg GAE g^−1^ DW)	TF (mg NAEg^−1^ DW)	DPPH (%)	FRAP (mmol g^−1^)
Variety									
BaklC	9.97 ± 0.60 e	6.50 ± 0.55 d	1.81 ± 0.06 b	352.1 ± 3.56 h	0.57 ± 0.06 e	6.18 ± 0.35 d	0.39 ± 0.04 d	83.1 ± 1.56 f	26.27 ± 0.95 f
Chba	11.14 ± 0.52 g	8.06 ± 0.45 e	1.86 ± 0.04 d	337.3 ± 4.78 g	0.47 ± 0.02 d	5.82 ± 0.78 d	0.39 ± 0.05 d	86.2 ± 1.05 g	25.20 ± 1.77 e
BaklS	9.45 ± 0.74 d	9.68 ± 0.43 g	1.97 ± 0.05 f	292.0 ± 2.87 d	0.67 ± 0.07 f	5.89 ± 0.55 d	0.38 ± 0.04 d	86.3 ± 0.98 g	26.32 ± 1.8 f
FKbM	11.32 ± 0.59 g	8.08 ± 0.65 f	2.37 ± 0.03 i	182.9 ± 2.87 b	0.29 ± 0.05 a	5.18 ± 0.66 c	0.37 ± 0.05 d	76.8 ± 0.88 e	20.67 ± 0.91 bc
FKbK	11.92 ± 0.45 h	14.05 ± 0.34 j	2.83 ± 0.04 j	158.0 ± 3.9 a	0.48 ± 0.07 d	4.89 ± 0.43 c	0.34 ± 0.02 c	77.3 ± 1.45 e	21.83 ± 0.98 d
Kna	8.63 ± 0.63 b	6.47 ± 0.36 c	1.83 ± 0.05 c	366.1 ± 4.87 i	0.43 ± 0.04 c	3.74 ± 0.23 a	0.33 ± 0.06 c	58.3 ± 1.64 b	20.88 ± 102 cd
Bel	7.52 ± 0.54 a	2.53 ± 0.56 a	1.67 ± 0.07 a	532.5 ± 3.02 k	0.41 ± 0.08 b	4.07 ± 0.56 ab	0.33 ± 0.02 c	60.0 ± 0.14 c	21.91 ± 0.88 d
SisS	11.12 ± 0.78 g	14.63 ± 0.69 k	2.09 ± 0.09 g	273.3 ± 4.76 c	0.76 ± 0.08 gh	3.90 ± 0.75 ab	0.33 ± 0.02 c	49.9 ± 0.22 a	20.62 ± 0.85 bc
Bka	9.09 ± 0.67 c	5.75 ± 0.98 b	1.83 ± 0.05 c	388.0 ± 4.67 j	0.77 ± 0.05 h	4.28 ± 0.45 b	0.26 ± 0.03 a	57.2 ± 1.32 b	19.60 ± 1.04 ab
SisC	10.70 ± 0.47 f	11.58 ± 0.45 h	2.11 ± 0.04 h	328.9 ± 3.89 e	0.75 ± 0.02 g	3.90 ± 0.32 ab	0.28 ± 0.04 ab	61.6 ± 1.68 d	19.45 ± 0.76 a
CGaz	9.89 ± 0.87 e	11.62 ± 0.67 i	1.88 ± 0.03 e	334.5 ± 5.96 f	0.68 ± 0.08 f	4.03 ± 0.22 ab	0.29 ± 0.02 b	61.0 ± 0.94 cd	20.03 ± 0.69 abc
Ripening stage									
S1	9.99 ± 0.56 a	8.92 ± 0.44 b	2.00 ± 0.03 a	318.89 ± 4.88 a	0.58 ± 0.08 b	3.35 ± 0.87 a	0.39 ± 0.06 c	59.28 ± 0.88 a	17.90 ± 0.40 a
S2	10.29 ± 0.86 b	8.91 ± 0.67 a	2.02 ± 0.02 b	325.85 ± 3.9 c	0.57 ± 0.04 b	4.45 ± 0.66 b	0.32 ± 0.08 b	70.76 ± 1.05 b	22.64 ± 0.55 b
S3	9.93 ± 0.77 a	9.15 ± 0.55 c	2.05 ± 0.04 c	322.37 ± 2.4 b	0.56 ± 0.05 a	6.36 ± 0.76 c	0.29 ± 0.03 a	76.72 ± 1.76 c	25.67 ± 0.76 c
Fruit position									
PII	9.20 ± 0.56 a	8.25 ± 0.48 a	1.50 ± 0.04 a	379.12 ± 3.43 b	0.52 ± 0.04 a	5.54 ± 0.98 b	0.35 ± 0.04 b	70.66 ±0.66 b	23.18 ± 0.66 b
PI	10.94 ± 0.66 b	9.74 ± 0.59 b	2.55 ±0.03 b	265.62 ± 3.76 a	0.62 ± 0.03 b	3.89 ± 0.88 a	0.32 ± 0.04 a	67.19 ± 0.76 a	20.96 ± 0.54 a

FW: fruit weight; FL: fruit length; Fth: fruit thickness; Y: yield; CAP: capsaicin, TPC: total phenolic content; TFC: total flavonoid content; DPPH: 1,1-diphenyl-2-picryl hydrazyl; FRAP: ferric reducing power assay; BaklC: Baklouti Chébika; Bel: Beldi; Chba: Chaabani; SisC: Sisseb Chébika; Bka: Bkalti; Kna: knaiss; BaklS: Baklouti Sbikha; SisS: Sisseb Sbikha; FkbM: Fort Menzeltemim; FkbK: Fort de korba; and CGaz: Corne de Gazelle. PI: lower fruit position plant; PII: upper fruit position; S1: green ripening stage; S2: orange ripening stage; and S3: red ripening stage. Values are the means of three different pepper samples (*n* = 3) ± standard deviation. Different letters (a–k) for the same parameter indicate significant differences among variety, fruit position, and ripening stage (*p* ≤ 0.05).

**Table 4 foods-12-03948-t004:** ‘Variety × ripening stage’ interaction of phytochemical traits for the studied traits in eleven Tunisian pepper varieties. Values are the means of three different pepper samples (*n* = 3) ± standard deviation.

Ripening Stages															
	CAP			TPC			TFC			DPPH			FRAP		
	S1	S2	S3	S1	S2	S3	S1	S2	S3	S1	S2	S3	S1	S2	S3
BaklC	0.77 ± 0.05	0.49 ± 0.07	0.76 ± 0.06	4.56 ± 0.42	5.81 ± 0.38	6.96 ± 0.31	0.45 ± 0.03	0.38 ± 0.04	0.35 ± 0.02	88.93 ± 3.05	89.78 ± 3.37	91.70 ± 3.36	25.94 ± 1.61	27.25 ± 1.34	27.99 ± 1.26
Chba	0.77 ± 0.06	0.77 ± 0.07	0.75 ± 0.04	4.53 ± 0.26	5.82 ± 0.29	7.11 ± 0.25	0.46 ± 0.04	0.37 ± 0.04	0.34 ± 0.02	88.93 ± 3.87	89.78 ± 3.80	92.77 ± 4.02	22.81 ± 1.08	25.97 ± 1.31	29.58 ± 1.38
BaklS	0.74 ± 0.04	0.76 ± 0.04	0.74 ± 0.04	4.58 ± 0.38	5.92 ± 0.33	7.17 ± 0.29	0.47 ± 0.04	0.35 ± 0.04	0.30 ± 0.03	89.77 ± 2.96	90.10 ± 3.62	93.62 ± 2.82	23.08 ± 0.87	27.76 ± 1.56	29.44 ± 0.82
FKbM	0.69 ± 0.07	0.66 ± 0.06	0.67 ± 0.05	3.51 ± 0.27	4.65 ± 0.24	7.19 ± 0.30	0.43 ± 0.03	0.35 ± 0.04	0.32 ± 0.05	70.24 ± 3.38	80.00 ± 4.89	91.51 ± 5.46	19.42 ± 1.72	19.35 ± 1.54	25.12 ± 1.30
FKbK	0.57 ± 0.07	0.56 ± 0.04	0.56 ± 0.06	3.48 ± 0.35	4.57 ± 0.33	6.64 ± 0.31	0.39 ± 0.03	0.34 ± 0.06	0.29 ± 0.03	68.25 ± 3.12	79.50 ± 4.52	92.73 ± 3.17	19.42 ± 1.57	21.52 ± 1.35	26.72 ± 1.40
Kna	0.48 ± 0.08	0.46 ± 0.04	0.47 ± 0.02	2.51 ± 0.03	3.38 ± 0.03	5.38 ± 0.33	0.39 ± 0.06	0.33 ± 0.04	0.28 ± 0.02	47.98 ± 2.78	57.62 ± 3.02	73.61 ± 2.64	15.25 ± 2.80	21.32 ± 1.03	24.97 ± 0.70
Bel	0.69 ± 0.05	0.67 ± 0.06	0.65 ± 0.04	2.29 ± 0.24	3.46 ± 0.34	6.03 ± 0.35	0.37 ± 0.04	0.33 ± 0.07	0.29 ± 0.03	50.08 ± 2.90	61.62 ± 3.03	73.18 ± 2.60	15.22 ± 1.09	21.27 ± 1.48	28.37 ± 1.30
SisS	0.33 ± 0.06	0.29 ± 0.04	0.27 ± 0.04	2.41 ± 0.27	3.81 ± 0.23	5.51 ± 0.25	0.36 ± 0.06	0.32 ± 0.02	0.30 ± 0.04	33.49 ± 3.13	57.43 ± 3.53	58.84 ± 3.67	14.19 ± 1.42	21.30 ± 1.14	26.42 ± 1.52
Bka	0.44 ± 0.06	0.50 ± 0.05	0.48 ± 0.05	2.92 ± 0.25	3.39 ± 0.23	6.09 ± 0.26	0.31 ± 0.06	0.25 ± 0.02	0.21 ± 0.04	45.46 ± 3.72	57.46 ± 3.07	68.90 ± 3.30	13.04 ± 1.18	21.64 ± 1.03	24.17 ± 0.90
SisC	0.45 ± 0.07	0.43 ± 0.06	0.41 ± 0.04	2.26 ± 0.04	3.43 ± 0.36	6.01 ± 0.37	0.34 ± 0.02	0.25 ± 0.05	0.24 ± 0.04	45.46 ± 3.59	69.52 ± 2.94	69.97 ± 2.50	12.64 ± 1.71	18.91 ± 1.35	26.80 ± 1.55
CGaz	0.43 ± 0.07	0.41 ± 0.04	0.37 ± 0.01	2.53 ± 0.30	3.74 ± 0.24	5.80 ± 0.27	0.34 ± 0.06	0.26 ± 0.05	0.25 ± 0.03	45.57 ± 2.67	68.56 ± 2.71	68.93 ± 3.79	15.97 ± 1.54	22.46 ± 1.17	21.66 ± 1.70

CAP: capsaicin (mg CAP g^−1^ DW); TPC: total phenolic content (mg GAE g^−1^ DW); TFC: total flavonoid content (mg NAE g^−1^ DW); DPPH: 1,1-diphenyl-2-picryl hydrazyl (%); FRAP: ferric reducing power assay (mmol g^−1^); PI: lower fruit position plant; PII: upper fruit position; S1: green ripening stage; S2: orange ripening stage; and S3: red ripening stage.

**Table 5 foods-12-03948-t005:** Fruit position × ripening stage interaction of phytochemical traits for the studied traits in eleven Tunisian pepper varieties. Values are the means of three different pepper samples (*n* = 3) ± standard deviation.

	CAP			TPC			TFC			DPPH			FRAP		
	S1	S2	S3	S1	S2	S3	S1	S2	S3	S1	S2	S3	S1	S2	S3
PI	0.63 ± 0.07	0.63 ± 0.04	0.61 ± 0.04	2.93 ± 0.33	3.99 ± 0.26	4.56 ± 0.31	0.39 ± 0.04	0.29 ± 0.04	0.27 ± 0.03	58.37 ± 2.91	69.07 ± 2.6	74.77 ± 2.46	17.11 ± 1.7	23.66 ± 1.29	24.26 ± 1.8
PII	0.52 ± 0.05	0.52 ± 0.05	0.50 ± 0.03	3.53 ± 0.30	4.91 ± 0.26	8.14 ± 0.30	0.38 ± 0.06	0.35 ± 0.08	0.27 ± 0.03	64.21 ± 2.4	68.99 ± 2.9	84.46 ± 2.71	18.79 ± 0.93	23.66 ± 1.29	28.66 ± 1.31

CAP: capsaicin (mg CAP g^−1^ DW); TPC: total phenolic content (mg GAE g^−1^ DW); TFC: total flavonoid content (mg NAE g^−1^ DW); DPPH: 1,1-diphenyl-2-picryl hydrazyl (%); FRAP: ferric reducing power assay (mmol g^−1^); PI: lower fruit position plant; PII: upper fruit position; S1: green ripening stage; S2: orange ripening stage; and S3: red ripening stage.

**Table 6 foods-12-03948-t006:** Correlation matrix of both agro-morphological and phytochemical traits collected from upper (PI) and lower (PII) fruit positions of plants of eleven Tunisian pepper varieties.

	PI									PII								
	FW	FL	Fth	Y	CAP	TPC	TFC	DPPH	FRAP	FW	FL	Fth	Y	CAP	TPC	TFC	DPPH	FRAP
FW	-									-								
FL	0.56 *	-								0.51 *	-							
FW	0.68 *	0.69 *	-							0.69 *	0.67 *	-						
Y	−0.62 *	−0.75 *	−0.69 *	-						−0.65 *	−0.71 *	−0.68 *	-					
CAP	0.56 *	0.58 *	0.65 *	−0.45	-					0.57 *	0.57 *	0.63 *	−0.47	-				
TPC	0.21	0.20	0.23	−0.55 *	0.73 *	-				0.22	0.20	0.25	−0.56 *	0.74 *	-			
TFC	0.19	0.15	0.19	−0.65 *	0.67 *	0.81 *	-			0.18	0.17	0.19	−0.67 *	0.68 *	0.82 *	-		
DPPH	0.13	0.17	0.22	−0.59 *	0.73 *	0.95 *	0.81 *	-		0.16	0.16	0.20	−0.56 *	0.75 *	0.94 *	0.84 *	-	
FRAP	0.18	0.15	0.23	−0.65 *	0.80 *	0.94 *	0.76 *	0.94 *	-	0.18	0.19	0.26	−0.65 *	0.82 *	0.92 *	0.75 *	0.93 *	-

FW: fruit weight (cm); FL: fruit length (cm); Fth: fruit thickness (mm); Y: yield (g plant^−1^); CAP: capsaicin (mg CAP g^−1^ DW); TPC: total phenolic content (mg GAE g^−1^ DW); TFC: total flavonoid content (mg NAEg^−1^ DW); DPPH: 1,1-diphenyl-2-picryl hydrazyl (%); FRAP: ferric reducing power assay (mmol g^−1^); and * indicates significance at *p* ≤ 0.05.

## Data Availability

Data are contained within the article.

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
