# Peer review of "Changes in Yield-Related Traits, Phytochemical Composition, and Antioxidant Activity of Pepper (Capsicum annuum) Depending on Its Variety, Fruit Position, and Ripening Stage"

_foods, 2023, doi:10.3390/foods12213948_

Round 1
Reviewer 1 Report
Comments and Suggestions for Authors
Dear Authors
The manuscript gives some idea about the changes in yield related traits, phytochemical composition, and antioxidant activities in pepper by accession, fruit position, and ripening stages. However, following points should be considered to enhance the quality of the manuscript:
1. There are several reports which give clear idea about the harvesting time dependent as well as ripening dependent variation in major bioactive compounds and antioxidant activities in pepper including. Some of them are https://doi.org/10.1016/j.foodchem.2022.133979, https://doi.org/10.1007/s13580-016-1008-6. Authors are encouraged to elaborate the introduction section.
2. What were the basic criteria for the separation of upper and lower fruit position?
3. How much fruit was collected from each of the plants and were those fruits good enough to represent sampling?
4. Were all the accessions orange color in middle ripening stage and red color in fully ripen stage?
5. The discussion is poor. I could not find supporting ideas about their results especially in the effect of ripening. Several reports are available elsewhere.
6. DMRT was performed separately among the accessions, ripening stage, and fruit position in table 2. However, it is not mentioned in the foot note. Furthermore, it will be better to use ANOVA rather than DMRT in fruit position as degree of freedom is 1 in that case.
In figure 1d, what is the Y axis? Furthermore, why the intensity of color in bar diagram is different among sub figures?
7. Line 376-377 what is the meaning of this sentence?
8. The reference section should be re-written as there are numerous errors throughout the section.
With Regards
Reviewer
Comments on the Quality of English LanguageMinor editing of English is required
Author Response
X
Question 1 :
There are several reports which give clear idea about the harvesting time dependent as well as ripening dependent variation in major bioactive compounds and antioxidant activities in pepper including. Some of them are https://doi.org/10.1016/j.foodchem.2022.133979, https://doi.org/10.1007/s13580-016-1008-6. Authors are encouraged to elaborate the introduction section.
Response 1
Response 1 :
Introduction was modified
What were the basic criteria for the separation of upper and lower fruit position ?
Response 2 :
Upper fruits are those located higher up on the plant at the 6th foliar stage, closer to the main stem or trunk, while lower fruits are positioned lower on the plant, closer to the ground at the second foliar stage.
Question 3 :
How fruit much collected from each of the plants and were those fruits good enough to represent sampling?
Response 3
The fruits selected for sampling should be free from visible defects, diseases, or damage. High-quality fruits are more representative : Fruits should be at an appropriate level of ripeness for sampling.
Question 4 :
Were all the accessions orange color in middle ripening stage and red color in fully ripen stage?
Response 4 :
Yes, all the accessions were orange in middle ripening stage and red color in fully ripen stage.
Question 5 :
The discussion is poor I could not find supporting ideas about their results especially in the effect of ripening . Several reports are available elsewhere :
- Response 5 :
Some sentence are added :
- Hamed, M.; Kalita, D.; Bartolo, M.E.; Jayanty, S.S. Capsaicinoids, Polyphenols and Antioxidant Activities of Capsicum annuum: Comparative Study of the Effect of Ripening Stage and Cooking Methods. Antioxidants2019, 8, 364.
https://doi.org/10.3390/antiox8090364
- Ye, Z., Shang, Z., Li, M., Zhang, X., Ren, H., Hu, X., & Yi, J. (2022). Effect of ripening and variety on the physiochemical quality and flavor of fermented Chinese chili pepper (Paojiao). Food Chemistry, 368, 130797. https://doi.org/10.1016/j.foodchem.2021.130797
- Ananthan R, Subhash K, Longvah T. Capsaicinoids, amino acid and fatty acid profiles in different fruit components of the world hottest Naga king chilli (Capsicum chinense Jacq). Food Chem. 2018 Jan 1;238:51-57. doi: 10.1016/j.foodchem.2016.12.073.
- Bhandari SR, Bashyal U, Lee Y-S (2017) Variations in Proximate Nutrients, Phytochemicals, and Antioxidant Activity of Field-cultivated Red Pepper Fruits at Different Harvest Times. Hortic. Environ. Biotechnol. 57(5):493-503. 2016. DOI 10.1007/s13580-016-1008-6
Question 6 :
DMRT was performed separately amoung the accessions, ripening stage and fruit position in table 2 : However, it is not mentioned in the foot note.
Further i twill be better to use ANOVA rather than DMRT in fruit position as degree of freedom is 1 in that case.
In 1D, what is the Y axis?
Response 6 :
Figure 1D were corrected.
Y is the yield parameter.
Question 7 :
Line 376-377 what is the meaning of sentence?
Response 7 :
The sentence was modified
Question 8
The reference section should be re-written as there are numerous errors
Response 8
The names of species are italicized

Reviewer 2 Report
Comments and Suggestions for Authors
The manuscript investigates an important topic – the dependence of some characteristics of chili pepper on 11 Tunisian accessions, 2 fruit positions and 3 ripening stages, using standard methods, and the results are summarized using statistical analysis. However, the results are not presented in the best way, so that the publication needs some revision and careful checking of the language.
Some recommendations:
- - The differences of the accessions are not presented in any way, a picture or description is recommended;
- - Table 1 is not sufficiently well explained and the authors may consider whether it would be better to be exchange it with Table 2;
- - Table 2 has several different formats that need to be corrected, the letters need to be better explained and placed after the corresponding numbers with or without a space for all numbers;
- - Both tables would be more suitable for supplementary information, rather than in the main text, e.g. Table 2 and Figure 1 provide identical information;
- - It is not clear why Fig. 1d lacks information about the y dimension and is different from 1a-1c. The information in the y-dimension must have the units represented;
- - In Table 3 some numbers are not well ordered and have different numbers after the decimal point;
- - Fig. 2 and Fig. 3 should be placed in the results section and their captures should be better defined. - - Fig. 2 is not of best quality;
-
Comments on the Quality of English Language- Term like “in other side”, “main and major” should be avoided.
Author Response
RESPONSES FOR REVIEWER 2
- - The differences of the accessions are not presented in any way, a picture or description is recommended;
- - Table 1 is not sufficiently well explained and the authors may consider whether it would be better to be exchange it with Table 2;
- - Table 2 has several different formats that need to be corrected, the letters need to be better explained and placed after the corresponding numbers with or without a space for all numbers;
Ok, the format of table 2 was corrected
In table 2, different letters (a–k) for the same parameter indicate significant differences among variety, fruit position, and ripening stage (P≤0.05)
- - Both tables would be more suitable for supplementary information, rather than in the main text, e.g. Table 2 and Figure 1 provide identical information;
Figure 1 is placed as supplementary information
- - It is not clear why Fig. 1d lacks information about the y dimension and is different from 1a-1c. The information in the y-dimension must have the units represented;
Figure 1 was completed and placed as supplementary infmatior
- - In Table 3 some numbers are not well ordered and have different numbers after the decimal point;
Format of table 3 is corrected
- - Fig. 2 and Fig. 3 should be placed in the results section and their captures should be better defined. - - Fig. 2 is not of best quality;
Ok, Figures 2 and 3 have been placed in the results section and extended
- Term like “in other side”, “main and major” should be avoided.
Both terms are moves and replaced

Reviewer 3 Report
Comments and Suggestions for Authors
This manuscript investigates the relationship between yield-related traits and active components of peppers with their variety, fruit position and ripening stage. A comprehensive analysis of the measured data revealed some regularities, and the results provide a reference for the later cultivation and processing of peppers with different characteristics. However, there are some issues in the manuscript need improvement.
These are some comments:
1. “accession” in this manuscript means the “variety”, if is, suggest change it to “variety” in the title and text.
2. In the abstract, “Variable responses to maturity stage and fruit position effects were obtained for all the biochemical CAP, TPC and TFC as well as the antioxidant activity estimated both by 2,2-diphenyl-1-picrylhydrazyl (DPPH) and ferrous reducing antioxidant power (FRAP) assays and for the total yield.”, “PCA analysis and cluster analysis showed three distinguishable groups”, What is the significance of these expressions in the abstract? There should be more statements of research findings in the abstract. The significance of the study, personally, I feel that it could be better to provide different bioactive materials for processing.
3. Line43, “minor” change to “bioactive”.
4. Under what conditions are the values of the indicators in Table 1, and does Table 1 only need to focus on the values in parentheses and how were they calculated?
5. In Table 2, in determining the effect of Accession on the various indicators, how was the fruit Ripening stage or Fruit position controlled, and what conditions were selected? How were the other 2 factors controlled in similar measurements of the effect of Ripening stage or Fruit position on various indicators?
6. In Figure 1, only the values in Figure 1a are duplicated, the measured values in the other Figures are not duplicated and no error line is seen? Figure 1d lacks vertical coordinates, add units for the vertical coordinates of 4 picture in Figure 1.
7. For “Accession x fruit position”, why only 4 indicators were chosen. For “Accession x ripening stage” and “Fruit position (FP) x ripening stage (RS)”, why choose these 5 indicators.
8. Latin names of species need to be italicized.
Comments on the Quality of English LanguageModerate editing of English language required
Author Response
RESPONSES FOR REVIEWER 3
Question 1
Accession in this manuscript means variety , if is suggest change into variety in the title and text.
Response 1
We agree the comment, accession changed into variety in all the text of manuscript
Question 2
In the abstract ‘’ Variable responses to maturity stage and fruit position effects were obtained for all the biochemical CAP, TPC and TFC as well as the antioxidant activity estimated both by 2,2-diphenyl-1-picrylhydrazyl (DPPH) and ferrous reducing antioxidant power (FRAP) assays and for the total yield.’’ And ‘’ PCA analysis and cluster analysis showed three distinguishable groups ‘’ what is the significance of theses expressions in the abstract There should be more statments of research finding in the abstract. The significance of the study personally, I feel that it could be better to provide different bioactive materials for processing.
We agree the comment and these sentences were modified
Question 3
Line 43 minor to bioactive
Response 3
We agree the comment, ‘minor’ was changed into ‘bioactive’
Question 4
Under what condition are the values of the indicators in table 1 AND DOES Table 1 only need to focus on the values in parentheses and how were calculated?
Response 4
The table was modified according the suggestion. How it was calculated is reported on text of table.
Question 5
In table 2 in determining the effect of accession on the various indicators, how was the fruit ripening stage or fruit position controlled and what condition were selected? How were the other 2 factors controlled in similar measurement of the effect of ripening stages or fruit position on various indicators
Response 5
The criteria for fruit position and the days after planting, were added in the materials and methods section exactly in Plant material and sample processing (L83-85)
Question 6
In figure 1, only the value in figure 1a are duplicated. The measured values in the figures are not duplicated and nor error line is seen? Figure 1d lacks vertical coodinates and units for the vertical coodinates of 4 picture in Figure 1 .
Response 6
Figure 1 was modified as requested
Question 7
For accession×fruit position, why only 4 indicators were chosen . For accession ripening stage and fruit position (FP) ripening stage (RS), why choose 5 indicators ?
Response 7 :
There was no significant interaction between AxFP, AxRS and FPxRS for all agro-morphological fruit traits. Interactions in relation to phytochemical traits were more significant. Fruit position and ripening stage play a major role, according to phytochemi-cal amounts rather than agro-morphological ones. Variety, fruit position, and ripening stage effects dominate all interactions (Table 1).
Question 8
Latin names of species need to be italicized
Response 8
Latin names of species are italicized

Reviewer 4 Report
Comments and Suggestions for Authors
The manuscript contains major issues that I point out below.
- Not a clear hypothesis for the study, the introduction is poor without a significant number of reports about the effect of preharvest factors and their interaction. The work is simply observing and recording.
- Substandard discussion.
- The last paragraph of the conclusions is not based on any of the presented data. The authors just report some climate data for the whole year, not specific dates of harvest (just a period of 2 months) and suddenly in conclusions, they explain some of the fundings based on temperature etc.
- Based on the ANOVA AxFP had no or limited significant effect on both agro morphological and phytochemical traits, but a total section is presented, I can’t understand why.
- Some of the graphs (e.g Fig. 1) has no statistical difference, I can’t understand why the authors choose to present it.
Minor
· Replace keywords that already exist in the title.
· Clarification with more details about the upper and lower part.
Comments on the Quality of English Language
English language of the ms requires checks and changes.
Author Response
RESPONSES FOR REVIEWER 4
- Not a clear hypothesis for the study, the introduction is poor without a significant number of reports about the effect of preharvest factors and their interaction. The work is simply observing and recording.
- Substandard discussion.
several paragraps were added to enhance the discussion
- The last paragraph of the conclusions is not based on any of the presented data. The authors just report some climate data for the whole year, not specific dates of harvest (just a period of 2 months) and suddenly in conclusions, they explain some of the fundings based on temperature etc.
Air temperature at harvest, fruit maturity and color of the eleven Tunisian chili pepper variety at each stage are added in table 1.
- Based on the ANOVA AxFP had no or limited significant effect on both agro morphological and phytochemical traits, but a total section is presented, I can’t understand why.
- Some of the graphs (e.g Fig. 1) has no statistical difference, I can’t understand why the authors choose to present it.
Figure 1 was changed and added as supplementary information
Minor
Replace keywords that already exist in the title.
Keywords that already exixt in the text were replaced
Clarification with more details about the upper and lower part.
Upper fruits are those located higher up on the plant at the 6th foliar stage, closer to the main stem or trunk, while lower fruits are positioned lower on the plant, closer to the ground at the second foliar stage.

Round 2
Reviewer 1 Report
Comments and Suggestions for Authors
No comments
Reviewer 2 Report
Comments and Suggestions for Authors
The manuscript is revised according to the reviewer’s recommendations and can be accepted for publication. However, I recommend checking for consistency rows 36, 71, 298, 334, 450 and in Table 4 the first row (BaklC 0.77±0.05; 0.49±0.07; 0.76±0.06).
Reviewer 3 Report
Comments and Suggestions for Authors
Accept in present form